# Synthesis and Electron Transporting Properties of Diblock Copolymers Consisting of Polyfluorene and Polystyrene

**DOI:** 10.3390/ma17112694

**Published:** 2024-06-02

**Authors:** Jin Cheng, Ruoyu Jiang, Yuhua Shan, Hong Sun, Shinji Kanehashi, Kenji Ogino

**Affiliations:** 1Department of Chemical Engineering and Pharmaceutical Engineering, Changzhou Vocational Institute of Engineering, Changzhou 213164, China; chengjin82@163.com (J.C.);; 2Jiangsu Province Key Laboratory of Fine Petrochemical Engineering, Changzhou University, Changzhou 213164, China; 3Graduate School of Bio-Applications and Systems Engineering, Tokyo University of Agriculture and Technology, 2-24-16 Nakacho, Koganei-shi 184-8588, Japan; kanehasi@cc.tuat.ac.jp; 4Zhejiang Fenghong New-Material Co., Ltd., Huzhou 313300, China; 5Institute of Global Innovation Research, Tokyo University of Agriculture and Technology, 2-24-16 Nakacho, Koganei-shi 184-8588, Japan

**Keywords:** electron mobility, crystallinity, diblock copolymers, thermal properties, photoluminescent stability

## Abstract

Poly(9,9-di-*n*-octylfluorene) (PFO) is a promising material for polymer light-emitting diodes (PLEDs) due to its advantageous properties. To enhance its electron transporting capabilities, diblock polymers were synthesized by attaching polystyrene (PSt) chains of varying lengths to one end of the PFO molecule. In a comparative study with PFO homopolymer, the diblock polymers maintained similar thermal properties, absorption spectra, and photoluminescent stability, while exhibiting slightly deeper lowest unoccupied molecular orbital (LUMO) levels and higher crystallinity. Notably, diblock polymers with shorter polystyrene blocks demonstrated higher electron mobility than the PFO homopolymer and diblock polymers with excessively long polystyrene blocks. These findings suggest that the optimal chain length of the polystyrene block is crucial for maximizing electron mobility, thus offering valuable insights for designing high-performance PLED materials.

## 1. Introduction

Polymer light-emitting diode (PLED) materials have been employed to convert electrical energy into light due to their numerous advantages, including mechanical flexibility, low cost, high thermal stability, uniform brightness, rapid response time, and wide viewing angles [1,2,3,4,5,6].

Among these PLED materials, poly(9,9-di-*n*-octylfluorene) (PFO) exhibits highly efficient photoluminescence, good solubility in common organic solvents, and broad emission wavelength in blue, green, and red spectral regions [7,8,9,10]. Moreover, an octyl side chain in nine positions enhances the color stability by depressing the undesired green emission [11,12]. It raises the electron mobility by inducing the desired *β*-phase formation [13,14]. Nevertheless, PFO also encounters many limitations when applied in PLED devices, such as material instability, low power efficiency, short lifetime, size- and flexibility-induced performance uncertainty, and insufficient device encapsulation [15].

In these disadvantages, one of the main hindrances to the commercialization of PFO was the low power efficiency, which was reported to originate from Shockley–Read–Hall electron-trap-assisted recombination and imbalance of hole transporting and relatively lower electron transporting [16,17]. To solve the adverse effects caused by the inferior electron transporting properties, various strategies were proposed to offer a more regular electron transporting path by planarizing backbone structure [13,14,18] or forming a microphase separation structure [15,19,20].

Based on these strategies of optimizing PFO structure in devices, non-fluorescent, electrically inert polystyrene (PSt) was introduced to one end of PFO to form a diblock copolymer.

As an easily prepared polymer, PSt has been widely applied in organic field effect transistors and photovoltaic devices as superior additives [21,22] or excellent attached building blocks [23] to enhance their properties. Furthermore, owing to the nonpolar nature of PSt, these modified conjugated polymers will retain their electronic performances because they can avoid introducing additional charge traps [24]. Our recent research has also discussed the introduction of PSt as a side chain in PFO, which promoted the crystalline properties of PFO and increased its electron-transporting properties [25]. 

In this study, a diblock polymer polyfluorene-block-polystyrene (PFO-*b*-PSt) with different PSt polymer chains was prepared, and electron-only (EO) transporting detection devices were fabricated through solution processing to investigate the electron transporting properties of the diblock polymer. The effect and its origin are revealed in the PSt diblock section on electron transporting properties of PFO by measuring thermal, optical, electrochemical, and crystalline properties.

## 2. Experimental

### 2.1. Materials

2-(7-bromo-9,9-di-*n*-octyl-fluoren-2-yl)-4,4,5,5-tetramethyl-1,3,2-dioxaborolane (compound **1**) and 2-(4-iodophenoxy)-1-[tert-butyldimethylsiloxy]ethane (compound **2**) were prepared according to the established procedures [26,27]. Dry tetrahydrofuran (THF) and other reagents were commercial products from FUJIFILM Wako Pure Chemical Corporation in Osaka, Japan. α-monocarboxylic polystyrenes (PSt-COOHs) with number-average molecular weights (*M*_n_) of 750, 1250, and 1800 were synthesized according to our earlier reports [28].

### 2.2. Synthesis of Hydroxyl End-Functionalized Polyfluorene (PFO-2)

Into a 100 mL flask, degassing toluene (17 mL) and compound **1** (1.77 g) were added, the mixture was deoxygenated by three freeze-and-thaw cycles with a liquid nitrogen bath, then 2N K_2_CO_3_ (30 mL) (after 0.5 h nitrogen bubbling handling), compound **2** (35.4 mg), and Pd(PPh_3_)_4_(0) (37 mg) were added and deoxygenated by three freeze-and-thaw cycles again. The mixture was stirred at 90 °C under a nitrogen atmosphere for 24 h. After the reaction, chloroform (100 mL) was added at room temperature, and the mixture was then washed with brine three times and dried with MgSO_4_. A total of 0.99 g of the yellow solid product tert-butyldimethylsiloxyethoxy end-capped PFO (PFO-1) was obtained with a yield of 56% after filtration, concentration, and reprecipitation in methanol. 

Under a nitrogen atmosphere, hydrofluoric acid/pyridine (HF/pyridine) (0.36 mL) was dropwise added to the solution of PFO-1 (0.99 g) and dry THF (50 mL). The mixture was stirred at room temperature for 24 h, followed by concentration via evaporation and reprecipitation in methanol to give 0.80 g of the yellow solid product PFO-2 (yield: 80%).

### 2.3. Synthesis of PFO-b-PSt

The mixture of PFO-2 (0.5 g, 0.05 mmol), PSt-COOH (0.5 mmol), dicyclohexylcarbodiimide (DCC) (1.78 g, 8.6 mmol), and 4-dimethylaminopyridine (DMAP) (1.06 g, 8.6 mmol) was placed in a 100 mL flask. Dry dichloromethane (60 mL) was added as a solvent under a nitrogen atmosphere. The mixture was stirred at room temperature for 24 h. After the reaction, the resulting solution was evaporated to remove the solvent, and acetone (60 mL) was added and stirred overnight to recover and crush the product. The filtrated solid was then purified via Soxhlet extraction with acetone for 48 h to give the final PFO-*b*-PSt sample. Characteristics of PFO-2 and PFO-*b*-PSts are listed in Table 1.

### 2.4. Characterizations

^1^H-NMR spectra were acquired using a JEOL ALPHA300 instrument at 300 MHz and 25 °C, with deuterated chloroform (CDCl_3_) as the solvent and tetramethylsilane as the internal standard. 

The *M*_n_ and polydispersity index (PDI) were evaluated using gel permeation chromatography (GPC) with a JASCO RI-2031 detector. The eluent was chloroform, flowing at 0.5 mL/min at room temperature, and calibration was performed using standard polystyrene samples. 

The glass transition temperature (*T*_g_) of the polymers was determined by differential scanning calorimetry (DSC) using a Rigaku DSC-8230 under a nitrogen atmosphere, with heating and cooling rates of 10 °C/min. 

Ultraviolet-visible (UV-vis) absorption spectra were recorded on a JASCO V-570 spectrophotometer, and photoluminescence (PL) spectra were obtained using a JASCO FP-6500 spectrophotometer with excitation at 380 nm. Films were prepared on 25 mm × 25 mm glass plates by spin-coating a chlorobenzene solution (40 mg/mL) at 1000 rpm for 60 s, filtered through a 0.45 µm membrane filter. The film fabrication process was similar to that used for EO devices on indium tin oxide (ITO) substrates.

Cyclic voltammetry (CV) measurements were performed using a Niko Keisoku Model NPGFZ-2501-A potentiostat/galvanostat. All measurements were conducted at room temperature in a typical three-electrode cell, with a glassy carbon electrode as the working electrode, an Ag/AgCl reference electrode, and a platinum wire counter electrode, at a scan rate of 0.1 V/s. Acetonitrile and 0.1 M tetrabutylammonium tetrafluoroborate (Bu_4_NBF_4_) were used as the solvent and supporting electrolyte. 

The crystal structure of the films on glass was investigated by grazing incidence wide-angle X-ray diffraction (GIWAXD) using a RIGAKU SmartLab X-ray diffractometer (Cu K_α_, *λ* = 1.5418 Å, 45 kV, 200 mA), with measurements taken from 3° to 30° at a step size of 0.02° and a scan speed of 1°/min in the out-of-plane configuration. The incident angle was set at 0.14°. Film and metal thicknesses were measured using a BRUKER Dektak XT-S stylus profiler.

### 2.5. EO Device Fabrication

Before fabricating the EO devices, glass slides with an ITO pattern (resistance of 10 Ω per square) were cleaned using an alkaline cleaner under sonication, followed by rinsing with deionized water. The substrates were then cleaned with 2-propanol under sonication, rinsed with fresh 2-propanol, and dried using nitrogen gas. The EO devices were constructed with a layer configuration of ITO/aluminum (Al) (50 nm)/polymer layer (150 nm)/lithium fluoride (LiF) (0.5 nm)/Al (100 nm). A 50 nm thick aluminum layer was deposited onto the ITO substrate in a vacuum at a pressure of ca. 3.0 × 10^−4^ Pa. As the cathode, a 0.5 nm thick LiF layer followed by a 100 nm thick Al layer was deposited on the polymer layer at the same pressure, with deposition rates of 0.01 Å/s for LiF and 3 Å/s for Al using tantalum and tungsten boats, respectively. The typical polymer area size was 6 mm². The current–voltage (I/V) characteristics of these devices were measured using a direct-current voltage and current source/monitor (KEITHLEY, 2400, Cleveland, OH, USA).

## 3. Results and Discussion

### 3.1. Synthesis and Characterization

As shown in Figure 1, the synthesis of PFO-1 was carried out via the Suzuki–Miyaura coupling reaction. Through the deprotection of PFO-1, a low steric hindrance PhOCH_2_CH_2_OH group terminating PFO-2 was prepared. Then, PFO-2 and PSt-COOH were linked to form the diblock copolymer PFO-*b*-PSt through a subsequent mild Steglich coupling reaction carried out at room temperature for only 24 h. By employing different *M*_n_ PSt-COOH prepared via atom transfer radical polymerization (ATRP), well-defined diblock copolymers with specified polymer chain lengths of PFO and PSt were produced.

Figure 1 shows the ^1^H-NMR spectra of the homopolymer and the diblock copolymers. With the incorporation of PSt of different chain lengths, characteristic peaks corresponding to PSt appear in the range of 6.20–7.20 ppm (denoted as red b). Based on the integration values of these characteristic peaks and the characteristic peaks of PFO (denoted as red a), the *M*_n_ of the PFO and PSt segments in each polymer can be calculated, as shown in Table 1.

### 3.2. Thermal Properties

As shown in Figure 2, the homopolymer (PFO-2) and diblock copolymers (PFO-*b*-PSt1, PFO-*b*-PSt2, and PFO-*b*-PSt3) showed a glass transition around 62~63 °C. *T*_g_ was barely changed by the incorporation of PSt in one endcap of PFO.

### 3.3. Optical Properties

The optical properties in both states are represented in Figure 3 and Table 2. In the UV-vis absorption spectra, all homo- and diblock copolymers showed similar *λ*_max_ about 384 nm in solution state (Figure 3a) and film state (Figure 3b).

Based on Figure 3b, it is evident that all homopolymers and block copolymers exhibit a red-shifted band around 433 nm, attributed to the formation of the *β*-phase [29]. According to the Lambert–Beer law [30], PFO-*b*-PSt2 demonstrates a more pronounced redshift (*λ*_max_ ≈ 433 nm) in the *β*-phase formation compared to other PFO-*b*-PSt variants (as detailed in Figure 3b′). This characteristic endows PFO-*b*-PSt2 with several attractive features, including higher charge-carrier mobility, enhanced current efficiency, and excellent spectral stability in PLEDs [31].

As depicted in Figure 3c,d, all polymers exhibited similar PL spectra in both solution and film states. This observation suggests that the attachment of PSt as a block segment has minimal influence on the PL properties of PFO. The PL spectra of all polymers displayed a main emission peak of around 420 (solution)/439 (film) nm, a shoulder peak approximately at 442 (solution)/468 (film) nm, and a long-wavelength emission peak around 475 (solution)/495 (film) nm. These peaks are associated with the π-π* transitions within the polyfluorene molecules, the microstructural features of the polyfluorene chains, and further microstructural characteristics of the polyfluorene chains, respectively [32,33].

### 3.4. Electrochemical Properties

The electrochemical properties of the polymers were investigated using CV. As shown in Table 2, the reduction process of PFO-*b*-PSts started at about −0.69~−0.68 V, lower than that of PFO-2 (−0.74 V), which can be attributed to the fact that the presence of PSt at one end of the mainchain causes a different aggregating environment around the PFO, inducing slightly different electrochemical features. On the other hand, the chain length of PSt has little effect on the reduction potential remaining around −0.69 V.

The lowest unoccupied molecular orbital (LUMO) levels for the polymers were calculated and listed in Table 2, and the LUMO levels of PFO-*b*-PSts were about −3.51 eV, deeper than that of the PFO homopolymer (−3.65 eV). It indicated that introducing PSt in one endcap of the polymer chain can facilitate the electron injection from the cathode to the polymer layer via deepening their LUMO levels.

### 3.5. Crystalline Structure Analysis

Figure 4 showed GIWAXD diffraction profiles in the out-of-plane geometry for thin films fabricated from the chlorobenzene solution. The diblock copolymers exhibited stronger peaks at ca. 2θ ≈ 20°, corresponding to π-π stacking distances of 4.2–4.4 Å, and showed crystalline patterns with distinct diffraction peaks [34,35]. As for the homopolymer PFO-2, the peak was relatively weaker, implying a nearly non-crystalline (amorphous) profile. The same phenomena happened at 2θ ≈ 7.5°, which might correspond to lamellar-patterned d-spacings [36].

Furthermore, it was found that both peaks at ca. 2θ ≈ 20° and 2θ ≈ 7.5° were stronger with shorter PSt as a blocked section, indicating the inhibition of crystalline formation by the presence of an overlong PSt chain. 

### 3.6. Electron Transporting Properties

The performance of the EO devices incorporating the PFO-2 homopolymer and PFO-b-PSt block copolymers with the structure ITO/Al/polymer/LiF/Al was assessed. Electron mobility in the space-charge-limited current (SCLC) region was derived from the I-V characteristic curves using Equation (1). This calculation considered a series resistance of 12 Ω and assumed that the built-in voltage was negligible due to the minor work function difference between the electrodes [37].
(1)$$J=\frac{9}{8}\varepsilon_{r}\varepsilon_{0}\mu_{e}\frac{V^{2}}{L^{3}}$$

In Equation (1), *J* represents the electron current density, *μ_e_* is the electron mobility, *ε_r_* is the relative permittivity of the material (3.5), *ε*_0_ is the permittivity of vacuum, *L* is the thickness of the active layer, and *V* is the voltage drop across the device.

Table 3 and Figure 5 show the electron mobility of devices fabricated from PFO-2 and PFO-*b*-PSts, with PFO-*b*-PSt2 exhibiting the highest electron mobility of 9.8 × 10^−6^ cm^2^/Vs. This is attributed to the relatively higher *β*-phase formation in PFO-*b*-PSt2 compared to other PFO-*b*-PSts, as well as the increased crystal formation in PFO-*b*-PSt1 and PFO-*b*-PSt2 compared to PFO-2 and PFO-*b*-PSt3.

## 4. Conclusions

This study successfully synthesized well-defined diblock copolymers of PFO-*b*-PSt with varying lengths of PSt polymer chains. The incorporation of PSt at one endcap of PFO did not significantly alter the *T*_g_s but influenced the optical, electrochemical, and crystalline properties of the copolymers. Notably, PFO-*b*-PSt2 exhibited the highest electron mobility, attributed to its pronounced *β*-phase formation and enhanced crystallinity.

These conclusions are similar to those we obtained through the graft copolymerization modification of PFO [25]. However, in contrast to graft copolymers, they exhibit stable blue light performance.

These findings underscore the importance of block copolymer design in tuning the electronic properties of polymers. Future research should explore the detailed mechanisms behind the structure–property relationships observed in these materials. Additionally, investigating other block copolymer compositions and their impact on device performance will be crucial for advancing the development of high-performance electronic materials. Further studies on the long-term stability and real-world applications of these materials will provide valuable insights for their practical implementation in electronic devices.

## Data Availability

Data are contained within the article.

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
