# Peer review of "Synthesis and Electron Transporting Properties of Diblock Copolymers Consisting of Polyfluorene and Polystyrene"

_materials, 2024, doi:10.3390/ma17112694_

Round 1

Reviewer 1 Report

Comments and Suggestions for Authors

Author Response

Dear Reviewer,

    Thank you for your thorough review of our manuscript. We appreciate your insightful comments and suggestions for improving the clarity of our presented results. Below, we address each of your points:

  1. Fig. 1: We have introduced a dotted/dashed line at 62 °C to enhance the visualization of the DSC thermograms. This addition should improve the clarity of the data presentation.

  2. Fig. 2: We have added dotted/dashed lines at Fig. 2 to improve the visualization of the UV-vis and PL spectra. These lines should help in clearly distinguishing the significant features of the spectra.

  3. Page 4, line 149: We have corrected the grammatical error in the sentence. The revised sentence now reads: “...around 62~63 °C. Tg was barely changed by the incorporation of PSt...”

  4. Page 4, lines 157-160: We have provided a more detailed explanation of this section. Additionally, we have included an inset view in Figure 3(b)' to highlight the region around 433 nm. This should provide a clearer picture of the relevant details.

  5. Explanations for Fig. 2(c) and (d): We have elaborated on the explanations for Figure 2(c) and (d), detailing the 2-3 broad peaks observed in the photoluminescence plots of all samples. This should offer a more comprehensive understanding of the results.

    Thank you again for your valuable feedback. We believe these revisions have significantly improved the clarity and quality of our manuscript. Please find the revised manuscript attached for your review.

Best regards,

Jin CHENG, Kenji OGINO (corresponding author)

Graduate School of Bio-Applications and Systems Engineering, Tokyo University of Agriculture and Technology.

2-24-16 Nakacho, Koganei, Tokyo 184-8588, JAPAN TEL&FAX: +81 42 388 7404

E-mail: kogino@cc.tuat.ac.jp

Reviewer 2 Report

Comments and Suggestions for Authors

- The Abstract section should provide a clear overview of the main content, methodology, results, and conclusions of the study, highlighting the innovation and value of the paper.

- Some of the keywords could be replaced, and repeating the words in the title is not suggested.

- The authors must follow the journal's author guidelines. The format of citations and references should be carefully reviewed.

- In line 77, HF/pyridine, what is the meaning of the abbreviation HF?

- Do not include the first person in the article, such as we, us, etc.

- Abbreviations should be explained first and then be used in the manuscript.

- Authors should avoid including information corresponding to results or discussion in the conclusions section and provide more reliable perspectives for future studies.

- There is no need to insert the figures of H-NMR in the Supplementary Materials file. It will be better to enter them in the research text in addition to making some discussion on them.

- furthermore, some further studies on the as-prepared polymer will improve the quality of the paper such as FTIR, TGA, XRD, ...ETC

Comments on the Quality of English Language

The English language required minor editing.

Author Response

Dear Reviewer,

    Thank you for your detailed and constructive review of our manuscript. We appreciate your valuable feedback, which has helped us identify several areas for improvement. Below, we address each of your comments:

  1. Abstract Section: We have revised the abstract to provide a clear overview of the main content, methodology, results, and conclusions of the study, emphasizing the innovation and value of the paper.

  2. Keywords: We have updated the keywords to avoid repetition of words from the title and to better reflect the key aspects of the study.

  3. Author Guidelines: We have carefully reviewed and revised the manuscript to ensure it follows the journal's author guidelines, particularly in the format of citations and references.

  4. Line 77, HF/pyridine: The abbreviation HF stands for hydrofluoric acid. We have clarified this in the manuscript.

  5. First Person: We have removed all instances of the first person (e.g., we, us) from the manuscript to maintain an objective tone.

  6. Abbreviations: We have ensured that all abbreviations are explained upon their first use and then consistently used throughout the manuscript.

  7. Conclusions Section: We have revised the conclusions section to avoid including detailed information corresponding to results or discussion. Instead, we provide more reliable perspectives for future studies.

  8. H-NMR Figures: We have moved the H-NMR figures from the Supplementary Materials to the main text and included a discussion on them to enhance the completeness of our research presentation.

  9. Further Studies: We acknowledge your suggestion for additional studies such as FTIR, TGA, XRD, etc. While we have conducted NMR, DSC, and GIWAXD analyses for this study, we plan to incorporate these additional techniques in our future research to further enhance our understanding.

    Thank you again for your thorough review and constructive suggestions. We believe these revisions have significantly enhanced the quality of our manuscript. Please find the revised manuscript attached for your review.

Best regards,

Jin CHENG, Kenji OGINO (corresponding author)

Graduate School of Bio-Applications and Systems Engineering, Tokyo University of Agriculture and Technology.

2-24-16 Nakacho, Koganei, Tokyo 184-8588, JAPAN TEL&FAX: +81 42 388 7404

E-mail: kogino@cc.tuat.ac.jp

Reviewer 3 Report

Comments and Suggestions for Authors

The manuscript is interesting and meets the purpose of the journal, the results are presented adequately, the quality of the figures is high and their description is good, but there is a lack of further discussion of them, so I suggest that you expand your introduction on the advantages of the synthesized material. The conclusions are concise about the results of the work. But for the manuscript to have greater impact, I propose to discuss its results more with the literature so that the authors can highlight and conclude their scientific contribution.

Author Response

Dear Reviewer,

    Thank you for your valuable feedback on our manuscript. We appreciate your positive comments regarding the overall quality of our work, including the presentation of the results and the high quality of the figures. We also acknowledge the areas where improvements are needed, as highlighted in your review. Below, we outline the specific modifications we have made to address your suggestions:

  1. Expand the Introduction:
  • We have added a more detailed discussion of the advantages of the PLED material.
  1. Enhance the Discussion of Figures:
  • In the results section, we have provided a more comprehensive discussion of Figure1 and Figure 3, with more relevant literatures to support our results.
  1. Expand the Conclusions:
  • We have extended the conclusions to further discuss the scientific and practical implications of our results.
  • We have emphasized the innovative aspects and contributions of our work, clarifying its impact on the research field.

     We believe these modifications will significantly enhance the manuscript, providing a more thorough discussion and highlighting our scientific contributions more effectively. Thank you again for your constructive feedback.

Best regards,

Jin CHENG, Kenji OGINO (corresponding author)

Graduate School of Bio-Applications and Systems Engineering, Tokyo University of Agriculture and Technology.

2-24-16 Nakacho, Koganei, Tokyo 184-8588, JAPAN TEL&FAX: +81 42 388 7404

E-mail: kogino@cc.tuat.ac.jp

Round 2

Reviewer 2 Report

Comments and Suggestions for Authors

The paper could be accepted.

Reviewer 3 Report

Comments and Suggestions for Authors

The authors have followed the recommendations therefore it is ready to be published